# Electromagnetically Stimuli-Responsive Nanoparticles-Based Systems for Biomedical Applications: Recent Advances and Future Perspectives

**DOI:** 10.3390/nano11040848

**Published:** 2021-03-26

**Authors:** Raffaele Longo, Giuliana Gorrasi, Liberata Guadagno

**Affiliations:** Department of Industrial Engineering, University of Salerno, Via Giovanni Paolo II, 132, 84084 Salerno, Italy; ggorrasi@unisa.it

**Keywords:** nanoparticle, stimuli-responsive, hyperthermia, pulsed electric field, electromagnetic stimuli, electrospinning, nanocomposite membrane, tunable properties

## Abstract

Nanoparticles (NPs) in the biomedical field are known for many decades as carriers for drugs that are used to overcome biological barriers and reduce drug doses to be administrated. Some types of NPs can interact with external stimuli, such as electromagnetic radiations, promoting interesting effects (e.g., hyperthermia) or even modifying the interactions between electromagnetic field and the biological system (e.g., electroporation). For these reasons, at present these nanomaterial applications are intensively studied, especially for drugs that manifest relevant side effects, for which it is necessary to find alternatives in order to reduce the effective dose. In this review, the main electromagnetic-induced effects are deeply analyzed, with a particular focus on the activation of hyperthermia and electroporation phenomena, showing the enhanced biological performance resulting from an engineered/tailored design of the nanoparticle characteristics. Moreover, the possibility of integrating these nanofillers in polymeric matrices (e.g., electrospun membranes) is described and discussed in light of promising applications resulting from new transdermal drug delivery systems with controllable morphology and release kinetics controlled by a suitable stimulation of the interacting systems (nanofiller and interacting cells).

## 1. Introduction

In the 1970s, Prof. Speiser’s research team began to develop a new system that allows for a better delivery of drugs within the human body. In that period of great effervescence and innovativeness in world research, the emerging use of the vaccines needed several injections to ascertain the correct absorption of the biological system. For this reason, the researchers started developing nanocapsules and nanoparticles (NPs) for drug delivery [1,2,3]. A nanoparticle for pharmaceutical purposes is a solid colloidal particle characterized by a dimensional range between 10 and 1 µm. In the first period, it was not possible to obtain very small NPs: Often the dimensions of the NPs were around 500–600 nm [4]. Several studies demonstrated that the properties and functionalities of the NPs are strictly dependent on their size, so numerous processes were rapidly developed to obtain NPs of much lower dimensions and shape (e.g., co-precipitation, microemulsion, hydrothermia, sonochemistry, thermal decomposition, etc.) [5]. Several relevant aspects need to be considered for the design of the nanoparticles. In the case of magnetic nanoparticles, for example, higher magnetic properties were found for smaller sizes, but it is worth noting that an excessive dimensional reduction of the particle size may cause problems in the phase of its elimination from the human body.

For example, Shapero et al. [6] proved that the silica NPs of different dimensions (from 50 nm up to 300 nm) are able to be uptaken by human cells, and the smaller the NPs are, the better the uptake efficiency is. Interestingly, studies proved that the cell uptake decreases for smaller dimensions than 50 nm [7,8], and they showed an optimum around 50 nm (this happened, for example, due to a more efficient receptor-mediated endocytosis) [7,9]. Figure 1 reports a scheme of different types of endocytosis phenomena [10].

Other studies demonstrated also that human cells are able to internalize several types of nanosystems also (e.g., biopolymer nano-carriers, liposomial nanovescicles system), even if the mechanism is not still clear [11].

The scientific community has found a deep interest in studying NPs, for different reasons. Several drugs are not able to pass some biological barriers, or their permeability through these barriers (e.g., cell membrane, blood-brain barrier, etc.) is limited and not suitable for producing the desired effects. In this context, the NPs constitute a good “carrier” able to overcome these boundaries, and they provide two main advantages. Firstly, using nanoparticles as carriers, it is possible to extend the range of drugs useful for planned purposes. Secondly, high efficiencies can be obtained with reduced amounts of drugs in the human body: This a very relevant aspect, especially for drugs with high cytotoxicity (e.g., chemotherapy). For antitumor therapies, in fact, the research is mainly focused on two directions: One is the development of new molecules [12,13,14,15], and the other is the development of completely different systems, such as nanoparticles, hybrid materials, and fibrous membranes, that are able to carry active molecules or fight against cancer [16,17,18,19]. This latter aspect has determined a spreading interest in nanoparticles for antitumor treatment and in the biomedical field. In Figure 2, the number of articles published per year on the application of nanoparticles in the biomedical field is reported according to ScienceDirect.

However, there are several types of NPs. Depending on the chemical nature/composition, the nanoparticles can be classified into four groups [20], as shown in Figure 3.

In this context, with a continuous need of innovative drug administration system, another interesting topic is the design of smart materials (SMs). SMs are generally stimuli- responsive systems that change their behavior when triggered. The activating agents can be, for example, enzymes, pH, ultrasounds, electromagnetic radiations, etc. [21,22,23,24,25,26,27].

In particular, electromagnetic radiations find a great variety of applications in the biomedical field, from diagnosis to therapy. Figure 4 illustrates some relevant applications of the wavelength ranges of electromagnetic radiations currently applied in the biomedical field [28,29,30,31,32,33].

The objective of this review is to present the main use of electromagnetically stimulable systems based on NP triggering. In particular, in the first section, NP functionalities and related results in biological environment are analyzed in depth, whereas in the second section, nanocomposite stimulable systems are presented and their features are explained on the basis of NP performance. An overview of the content is reported in Figure 5 and in Table 1.

In particular, the first part (Section 2, Section 3 and Section 4) is mainly focused on stimulable NPs, among which are magnetic and metallic NPs. Magnetic NPs can be easily urged by a magnetic field and, moreover, can be also led to the interest area that exploits the magnetic force. For this purpose, mainly magnetite (Fe_3_O_4_) and maghemite (γ-Fe_2_O_4_) are used. By exploiting their interaction with an alternate magnetic field (AMF), the stimulation leads to a temperature increase. This phenomenon is called hyperthermia, and in a certain range (around 41–47 °C) it is very effective for the treatment of many carcinomas. Similar effects are determined by metallic NPs, which can be urged by near-infrared (NIR) radiation, which stimulates the NP, leading to a temperature increase as well.

Concerning this last type of NPs, the potentiality of the pulsed electric field (PEF), which is actually clinically used for treatments of external solid tumors, was discussed in some studies. In this review, an interesting overview on the recent results with the combination of NPs and PEF is reported, which opens the way toward very interesting perspectives.

It is worth noting that the key for designing an effective smart NM, once the is material chosen, is to choose appropriately the stimulus, since it will be the way to interact with the NP.

The second part (Section 5) focuses on the biomedical applications of electrospun membranes composed of micro/nanofibers that contain integrated promising functionalities resulting from the abovementioned NPs embedded in the hosting fibers. This final analysis is carried out in terms of performance in the stimuli-responsive efficiency.

## 2. Magnetic Nanoparticles

### 2.1. Activation of an External Magnetic Field

A magnetic NP is a type of particle that manifests magnetic properties. In order to understand how to optimize the results, it is crucial to study the most important characteristics of this kind of particles. The magnetic nanoparticles that are mainly analyzed are iron oxides (usually magnetite). Magnetite has ferromagnetic properties: For this reason, once it has been magnetized through an external magnetic field, even if the external field stops, the magnetite works like a magnet. If the dimensions of the NP are small enough, the NP is not ferromagnetic but superparamagnetic. This happens when the dimensions of the particle are comparable to the ones of a single magnetic domain [34]. This dimension for the magnetite should be below 76 nm [35]. This phenomenon may happen for each NP that shows reticular energy similar to the energy necessary to invert the direction of the magnetic moment. The time necessary for the inversion of the magnetic moment direction is known as the Néel relaxation time. If this time is lower than the time necessary to test the measure of the magnetic moment, the measure will report a value of 0. For this reason, the material may seem paramagnetic, but it will show a magnetic susceptibility like a ferromagnetic material does. However, there are differences between a ferromagnetic nanoparticle and a superparamagnetic one, especially regarding the potential applications. If an external magnetic field would urge a ferromagnetic NP, it would reach the condition of magnetization saturation (B_SAT_) quickly, independent of whether the NP is superparamagnetic or ferromagnetic. Once the magnetic field is stopped, the nanoparticle loses the magnetization. However, it is not able to return to the value of 0 (the start point) of the magnetic field. The ferromagnetic materials store their history. The magnetic field that stores it without an external field is called magnetization residual (B_RES_). To force the magnetization to come back to 0, it is necessary to use another external field of a different direction. The point for which the magnetic field of the ferromagnetic NP is 0 is called coercivity (H_COE_). Obviously, it is possible to obtain another magnetization of the material with this external field, whereby it can reach another B_SAT_ condition. To reach the first magnetic saturation, it is necessary to reach another point of coercivity: The route of the diagram shows a hysteresis. The route for the superparamagnetic materials is different: In fact, they do not show any magnetic residual field and therefore no hysteresis [36]. Figure 6 reports the route of magnetization for a ferromagnetic and a superparamagnetic material.

The energy lost by the ferromagnetic materials because of the hysteresis is released in the surrounding environment as heat. For this reason, the magnetic NPs often are triggered by AMF to increase the average temperature of the interested area (located hyperthermia) [37]. The main parameters of the material for these NPs are the saturation magnetization, coercivity, and residual magnetization [38].

Although there is no hysteresis for superparamagnetic materials, it is possible to notice that in applying an external magnetic field, there is an increase of temperature of the NP. This happens because of the Néel relaxation time (NRT) and the Brownian relaxation time (BRT). Néel relaxation, as cited previously, is the phenomenon of the NPs able to invert their magnetic moments overcoming an energy barrier. When this happens, there is heat formation by the NP, which will affect the surrounding environment. The equation that describes the NRT is reported in Equation (1):(1)$$\tau_{N}=t_{0}*\exp\left(\frac{K*V}{k_{b}*T}\right)$$
where *t*_0_ is generally 10^−9^ s, *K* is the anisotropy constant, *V* is the NP volume, *k_b_* is the Boltzmann constant, and *T* is the temperature. It is evident that the smaller the particle is, the shorter the Néel relaxation time. As regard the BRT, it is related to the interaction of the nanoparticle with the medium. Equation (2) presents the model of the BRT.
(2)$$\tau_{B}=\frac{3*\mu *t_{0}*V}{k_{b}*T}$$

There is only one new term, µ, the viscosity of the system. Both BRT and NRT are strictly related to the dimensions of the particle, but Néel relaxation is related by an exponential form. This is the reason for which usually the assessments on the nanoparticles are done only referring to the NR. To obtain a quantitative measure of the heat dissipation, it is possible to use Equation (3):(3)$$P=\mu_{0}*\nu *f*H^{2}$$
where *P* is the heat dissipation, *f* is the frequency of the external magnetic field, *H* is the strength of the external magnetic field, µ_0_ is the magnetic field constant, and ν is the imaginary part of the magnetic susceptibility [39]. For this reason, it is necessary to carefully choose the material and the size: Several studies showed the dependence of the heating rate on the size of NPs [40]. Rosensweig demonstrated that there was a maximum rate that was independent of the external magnetic field. Moreover, it was shown that the magnetite, which did not show a monotone behavior, showed a maximum for the heating rate around 7 nm. However, studies confirm (in an intuitive way) that it is possible to enhance the hyperthermia effect by increasing the amplitude of the external magnetic field and the frequency.

### 2.2. Physicochemical Properties and Choice of Coating for Biomedical Application

Several studies reported that the NP design needs to be accurately chosen in a way that also depend on the physicochemical properties (e.g., size, surface charges) [41]. The size of the NPs strongly affects their distribution in the biological environment: In the case of injection, to make sure that the NPs would not be removed by the kidneys, their dimensions need to be more than 50 nm. Moreover, for dimensions more than 200 nm, the NPs are removed by the spleen and the liver. For this reason, the medicine establishes a range that is 50–200 nm for systematic delivery NPs [41,42]. A similar behavior is obtained in the case of inhalation: Braakhuis et al. [43] demonstrated that NPs above 60 nm accumulate in the lungs, whereas below 34 nm they accumulate in the alveoli. In addition, the uptake efficiency can be sensitively affected by the dimensions. For example, receptor-mediated internalization shows a higher uptake efficiency for 50 nm NPs, which seems to be the compromise between the curvature of NPs and the binding rigidity. Moreover, by appropriately designing the NPs’ physicochemical characteristics, it is often possible to affect the cytotoxic effects: For this reason, some of these are considered as essential information for toxicity studies (e.g., size, surface area, agglomeration state, composition, concentration) [44].

On the other hand, other parameters such as the surface charge (controllable, for example, by changing the NP coating [45]) usually are accurately chosen for controlling the uptake efficiency [46]. Several studies reported for example that positively charged NPs are more easily uptaken than neutral and negatively charged NPs [47,48,49], and this behavior strongly depends on the interaction between the membrane and the NPs. Table 2 reports the main effects of the physicochemical properties on biological variables.

However, to modify the physicochemical properties of NPs and their effect (e.g., biodistribution), it is possible to vary the architecture of NPs, creating heterogeneous NPs (e.g., Janus type) [55,56]. Due to the heterogeneity of NPs, for example, it is possible to obtain NPs with two faces that present different characteristics (e.g., metal and polymer). These NPs manifest interesting properties for NP solubility in complex systems and allow for dual-phase release kinetics, theranostics, and drug targeting [57,58,59,60].

The magnetic NPs generally are delivered in biological systems via two ways: direct injection, in which the magnetic NPs are directly injected in the tumor, or via systematic delivery, in which the NPs are injected intravenously in order to be delivered in the interested zone [36].

Direct delivery is generally preferable; it is possible to be quite sure of the concentration of NPs delivered in the tumor. However, the problem is that the tumor may not be accessible for direct injection in situ. In these cases, it is preferable to use systematic delivery: this works mainly due to the enhanced permeability and retention effect (EPR), which enhances the absorption of nanoparticles in the tumor tissue rather than the healthy one. This effect is widely studied and, even if not all the reasons are clear, the main reason is probably due to the leaky vascular system in the tumor tissue that promotes the permanence of the NPs in that zone [61]. In the case where NPs are accumulated in a certain zone only because of the physiology and the anatomy of the system, passive targeting occurs [62].

However, to enhance the selectivity of the delivery (especially for the systematic ones), the NPs can be functionalized to perform active targeting [62]. By coating the nanoparticles with agents (such as antibodies) that have a chemical affinity to the target cells, it is possible to selectively bind the NPs to the cells [63]. Thus, in coating the magnetic NPs, it is possible to obtain core-shell NPs. It is possible to use them to protect the NP from oxidation and to avoid the agglomeration phenomenon [64], thus preserving the magnetic properties [5]. For example, Sarno et al. [65] in 2017 analyzed how different types of coating affect NP properties. For three different types of coating (oleic acid, citric acid, and lipase-citric acid), the magnetic tests showed that the main parameters, such as the saturation magnetization and the coercivity, were found very similar, proving that the coating molecules slightly affects the magnetic properties. However, this strategy is able to enhance the EPR effect and the uptake by the cells as well. The efficiency of the NPs was demonstrated to change depending on the coating, which modifies magnetic properties (such as the coercivity), but some NPs, such as oleic acid-coated NPs (OANPs), were evaluated to have a higher heating rate than the uncoated ones, where the effect may be due to the relaxations losses [66]. Figure 7 presents a magnetite NP functionalized with oleic acid.

The coating agents usually used are polymers and polysaccharide (such as polyvinyl-alcohol [67], polyacrylic acid [68], dextran [69], chitosan [70]), or agents such as oleic acid [71] and citric acid [72]. It is necessary to choose the coating accurately because it can directly or indirectly affect the cell uptake. The balance between hydrophilicity and hydrophobicity for the NPs is particularly challenging. In fact, on the one hand, the hydrophilicity allows for a good dispersion in serum or in water, thereby avoiding aggregation. For water dispersibility, generally it is possible to attach hydrophilic functional groups to the ligands (e.g., carboxylic acids, sulfonic acids, PEG, etc.) [73]. On the other hand, the hydrophobicity enhances the interactions among the NPs and the cell membrane and therefore their uptake [74,75]. Given the fact that surface charge affects the cell uptake (generally positive surface charge increases the uptake), the use of cationic ligands was one of the main ways used to overcome the cell membrane. However, now the research is also going toward the development of cutting-edge NPs with hydrophobic/hydrophilic faces (Janus type) or hydrophilic/hydrophobic triggerable properties [73].

As explained before, it is possible to use agents able to target a certain type of cells: For example, Pala’s group [76] produced a dextran-coated superparamagnetic nanoparticle with HER2 aptamer (HNP) and tested these NPs to cell lines that express the HER2 receptor and those that do not express the HER2 receptor. It was possible to notice that the HNPs were much more specific than the NPs without the HER2 aptamer. The second reason is related to the changes in the electromagnetic and physicochemical properties: As anticipated before, the coating of the NPs would affect their surface charge, and uptake of NPs by cells is strictly dependent on this [77,78]. Ayala et al. [78] synthesized iron oxide NPs with different values of substitution of carboxymethyl-substituted dextran, and they monitored the variations of the surface charge. Through the evaluation of different uptakes of these NPs by the colon cancer cells, they found that the best uptake was manifested by the most charged nanoparticles. A different approach to improve the bioavailability of magnetic nanoparticles (mean size of 6.5 ± 3.0 nm) was the fabrication of nano-capsules with a shell of poly-lactic-co-glycolic acid (PLGA) or polylactic acid (PLA), further covered by carboxybetaine-functionalized chitosan by using dense gas technology. These systems were responsive to an external magnetic field that released an in vitro loaded fluorescent payload with a remote on/off control, achieved with alternating magnetic field [79].

In synthesis, on the one hand, the coating affect the uptake and cytotoxicity due to changes of relevant parameters (e.g., surface charge) [49]; on the other hand, it can be exploited to improve the selectivity of the NPs and promote active targeting.

### 2.3. Effectiveness of Hyperthermia

Since the first studies, nanoparticles seemed to be promising for application in drug delivery. Magnetic hyperthermia can cause protein denaturation and DNA damage of the cells. It can enhance the immune response and the drug perfusion within the tumor [80,81]. Cell death via magnetic hyperthermia can be due to apoptosis or necrosis. However triggering cell death via apoptosis is preferred than necrosis since necrosis can provoke inflammation or metastasis [82].

In 1983, Widder et al. [83] produced magnetically responsive microspheres containing doxorubicin (DOX) hydrochloride as treatment of Yoshida sarcoma tumors in rats. Over 77% of the rats showed total remission of the tumor, and the others showed a significant decrease in the tumor mass. For the animals non-treated in the same way (using free DOX and placebo microspheres), there was a relevant increase of the tumor mass with successive death. In this experiment, the magnetic field was generated by a permanent bipolar magnet, as chemotherapy was not combined with hyperthermia yet. The following studies with DOX bounded on magnetite NPs showed how the results improved if the NPs were exploited to generate heat. Ha et al. [84] synthesized superparamagnetic NPs with an alginate shell and DOX. Moreover, a folate factor was added on the surface of the cell to enhance the entrance and the permanence of the NPs in the system. Due to the external alternate magnetic field, the NPs generated heat, and by in vivo tests a promising decrease of the tumor size in the system was proven. In addition, studies on other chemotherapy agents were proven to be more effective: Petryk et al. [85] proved that it was possible to reduce the amount of cisplatinum (CP) necessary than the free CP or the CP with traditional heating techniques. In this case, the NPs were inserted after the chemotherapy treatment in mice for MTG-B flank tumors, and a significant loss of the tumor size compared to the initial one was found.

This enhancement of the effectiveness of hyperthermia in combination with chemotherapy is due to several factors. First of all, it has a beneficial effect on enhancing the immune response of the biological system. Secondly, when combined with chemotherapy, it shows an enhancement of the transport phenomena (that of the drug uptake and of the flux of anticancer drug concentration), which leads to major DNA damage of the cancer cells and a reduction of the radical oxygen detoxification [86].

Recently, doping magnetic NPs with small percentages of lanthanides has become an interesting pathway in cancer treatment [87]. Firstly, it is possible to produce NPs with higher magnetization properties. For example, in 2020 Kowalik et al. [88] produced iron oxide NPs with different percentages of yttrium (from 0.1% to 10%), and they reported differences in terms of the magnetic properties (hence heating capability) and cytotoxicity of the systems. Interestingly, an increase of over 60% of the heating capability was recorded with a 0.1% inclusion of yttrium in the system, which probably affected the crystalline structure and in turn, indirectly, the magnetic characteristics. Moreover, lanthanides can be exploited for their characteristics as radionuclides, leading to interesting horizons also in the diagnosis [89] or for antitumor treatment where the lanthanides is used as the internal radio emitter.

Even if recent magnetic studies are now well investigating the treatment of several type of cancers [80] (e.g., epidermoid [90], prostate [91], breast [92]), the use of magnetic nanoparticles find great application for the treatment of brain cancer; this happens mainly because most of the active substances are not able to overcome the blood–brain barrier (BBB), which safeguards the brain from the substances that may circulate in the blood. For this reason, it is very challenging to find delivery systems able to overcome the BBB [93]. In 2015, Dan et al. [94] studied the effect on the permeability of iron oxide NPs stimulated by AMF-induced hyperthermia through the BBB. In this study, citrate-coated iron oxide NPs were synthesized, and crosslinked nanoassemblies loaded with iron oxide NPs were successfully obtained. The in vitro tests were done at 37 °C and 43 °C, confronting the results with the local hyperthermia caused by AMF. This work showed that induced hyperthermia with magnetic NPs can be effective to overcome the BBB by accurately choosing the functionalization of the NPs and that the results can be very different for different cell lines. In vivo studies for the treatment of glioma were performed by Yanase et al. [95], who showed how with three applications of AMF (one each 24 h), 87.5% of the tumor had a reduction. Moreover, the distribution of NPs stimulated through AMF in the tumor tissue was homogeneous, whereas in the test without AMF the distribution of NPs was more heterogeneous. Encouraging in vivo results were also obtained by Jordan et al. [96]. They proved that a superparamagnetic iron oxide NP with aminosilane coating, after the exposure to AMF, guaranteed a decrease in cell proliferation and an enhancement of the survival rate for animals affected by malignant glioma.

Given these performance, clinical trials were performed for magnetic hyperthermia treatment of glioblastoma. The NPs used are aminosilane-coated superparamagnetic iron oxide NPs dispersed in water (NanoTherm), with a magnetic field frequency of 100 kHz, an intensity of 0–15 kA/m (NanoActivator) [97]. Comparing the results obtained for conventional treatments, the median survival rate was increased from 12.1 months for patients treated only with radiotherapy (median age 57 years old) and 14.6 months for patients treated with radiotherapy and chemotherapy (median age 56 years old) [98] to 23.2 months for the 59 patients tested (median age 56 years old) with magnetic hyperthermia and radiotherapy [99,100].

## 3. Metallic Nanoparticles

### 3.1. Activation via Near-Infrared Radiation (NIR)

Metallic NPs are made, for example, of gold, copper or silver, and are characterized by free-moving electrons. The quasi-particle formed by interactions of these moving electrons is called a plasmon. The plasmons on the surface of the materials are known as surface plasmons, and they are exposed to the surrounding environment and therefore to external radiation. If the metallic NPs are smaller than the wavelength of the incident radiation, a phenomenon known as localized surface plasmon (LSP) can occur, which is responsible for the oscillation of the plasmon with a defined frequency [101]. The oscillation that the NP gives back depends on the incident radiation. It has been noted that the highest oscillation amplitude given for a specific frequency is known as the local surface plasmon resonance [102]. As reported in Equation (4), the whole extinction cross section (*C_ext_*) is generally due to two contributions: the scattering (*C_sca_*) and the absorption (*C_abs_*) [103].
(4)$$C_{ext}=C_{abs}\left(\lambda \right)+C_{sca}\left(\lambda \right)$$

Considering spherical NPs with dimensions lower than 20 nm, it is possible to theoretically obtain the extinction cross-section value by Mie’s solution of Maxwell’s relations [104].
(5)$$C_{ext}=\frac{24*\pi^{2}*R^{3}*\varepsilon_{m}^{\frac{3}{2}}*N}{\lambda *\ln\left(10\right)}*\frac{\varepsilon_{i}}{{\left(\varepsilon_{r}+\chi *\varepsilon_{m}\right)}^{2}+\;\varepsilon_{i}^{2}}$$
(6)$$\varepsilon =\;\varepsilon_{r}+i*\varepsilon_{i}$$
where *R* is the radius of the NP, *ε_m_* is the dielectric constant of the medium, λ is the wavelength of the incident radiation, *N* is the electron density, ε is the complex dielectric constant of the NP, and *χ* is a factor describing the shape of the NP. In Figure 8, a metallic NP surface plasmon resonance induced by external radiation is represented.

Equation (5) highlights the point that the material’s response strongly depends on the size of the NP, the shape, and the dielectric environment [105]. It is proven that at low dimensions (≈20 nm), the extinction cross section is almost completely due to the absorption. By increasing the dimensions, the magnitude of the scattering phenomena starts to be relevant, and by nanospheres of 80 nm, the phenomenon starts to be comparable [106]. However, the NP shape is a significant parameter as well: Increasing the aspect ratio, the magnitude of the scattering and the absorption is equal at a much lower volume of the NP. Moreover, for these NPs, it is easier to modify the extinction cross section since there are two parameters by which it is possible to act, i.e., the diameter and the aspect ratio [106]. For metal NPs such as Au and Ag, the resonance frequency is in the near-infrared range [102], but it is interesting to notice a particular effect of the radiation: the radiation absorbed is dissipated by the material via heat release. NIR radiation in a range of 650–1350 nm can penetrate safely in depth in human tissue [107]. This is interesting because it means that by suitably choosing the NP size and type of surface functionalization, it is possible to enhance and control the heat generated in the tissue that causes hyperthermia [108]. To obtain the LSPR in the NIR region, nanoshells are often produced with a core of SiO_2_ and the outer layer of Au. Modifying the thickness of the outer layer, it is possible to vary the resonance frequency [109]. In Figure 9, a difference of the heat generated for various sizes of Ag-NPs under a 532 nm wavelength radiation through photothermal lens spectroscopy is shown [103].

The behavior of the graph in Figure 9 clearly evidences how the size of the NPs for a defined wavelength also affects the photothermal effect.

In synthesis, a suitable NIR-responsive NP should show a relevant absorption in the NIR region, which can be converted in heat. The most used materials in this sense are noble metal nanomaterials because of the strong plasmon resonance, even if other type of materials are now also gaining interest in being studied (e.g., graphene-based materials) [110,111].

### 3.2. Use of Gold Nanoparticles with NIR Radiation for Biomedical Application

Au-NPs are widely studied and are one of the most interesting metallic NPs. This is due to their low toxicity [112] and cytotoxicity [113]; these characteristics highlight a promising potentiality for their use for biomedical purposes. However, to enhance their compatibility with biological molecules and to use the NPs as drug carriers, they often need to be functionalized. Sulfur has a strong affinity to gold, and for this reason, thiols are often used to functionalize the Au-NP surface, which organizes themselves regularly on the NP surface, forming a self-assembled monolayer (SAM) [108,114]. The choice of the thiol is very important. The sulfur forms a bind with Au, but the rest of the organic molecule is free to react, as shown in Figure 10. For example, if the organic molecule has a carboxyl group [115], it would be reactive toward amine and therefore proteins [116]. In addition, the polyethylene-glycol (PEG) is often used to functionalize the Au-NPs surface, given its capacity to be bounded with molecules such as biotin, lactose, etc. [117,118]. Depending on the type of ligand, the NPs manifest different properties and possible applications. In 2008, Lee et al. [119] used Au-NPs functionalized by amine as carriers for intracellular delivery of small interfering RNA (siRNA) conjugated with PEG for treatment of prostate cancer. In this case, the gold has the function of enhancing the internalization of siRNA in the cell lines. By choosing the ligand accurately, it is even possible to do nuclear targeting [120]. In 2003, Tkachenko et al. [121] produced Au-NPs able to enhance a selective nuclear uptake of peptide-Au-NPs for liver carcinoma cells.

The dimensions obtained for these functionalized NPs can be very different depending on the ligands and on the original dimensions of the NPs. For example, for the nanoshells used under NIR radiation, the dimensions are often around 150 nm [122]. For this reason, they are suitable to be administrated not only via direct injection but also via systematic delivery, which permits them to reach the tumor site due to the EPR effect. The hyperthermia caused by NIR radiation has several physiological effects on the cells. Firstly, the cancer’s vessels and cell membrane permeability increase due to the rise of temperature, which causes a more efficient drug uptake. However, there are also other effects directly inside the cells: The temperature increase also causes protein denaturation and DNA damage, as detected in the case of magnetic hyperthermia [122]. This may lead to cell death, which can be of two types: necrosis or apoptosis. Even if the necrosis causes tumor cell death, this may cause the release of damaged biological material in the surrounding environment that may degenerate, in turn causing inflammatory and immunogenic responses, whereas apoptosis leads to tumor cell death that discourages inflammation [123]. For this reason, it is crucial to choose the parameters and material characteristics correctly in order to obtain the desired heating, thus to control the system response [122]. For this concern, research is developed toward the understanding and the definition of the physical parameters of the NPs in order to control the effects as much as possible. Carrillo-Cazares et al. [105], for example, determined the optical properties of Au-NPs when they are inserted in liver and colon tissue, and they focused on the differences in the temperature recorded between the two different environments. As with the case of magnetic hyperthermia, NIR radiation hyperthermia found scientific interest in the treatment of brain tumors. Bernardi et al. [124] produced gold-silica nanoshells for in vitro treatment of medulloblastoma and glioma. The nanoshells were functionalized with antibodies to interact with characteristic receptors present on the cell membranes, in order to enhance cell targeting. Results showed that for cell lines with receptors overexpressed, the targeting was effective, and in this way, it was possible to continue the treatment of photothermal ablation. Similarly, Carpin et al. [125] studied NIR hyperthermia of Au-NPs causing photothermal ablation for trastuzumab-resistant breast cancer cells. The results showed that this treatment was effective even for cells characterized by resistance to drugs. Combining drugs with photothermal treatment, however, is now one of the fields that are more promising. You et al. [126] produced Au-NPs coated with PEG and loaded with doxorubicin (DOX) for the treatment of breast and ovarian cancer in vivo. Under NIR radiation, the cell uptake of DOX increased for all the cell lines and showed a relevant decrease of the tumor size after more than 20 days by the treatment compared to the same treatment without NIR radiation. Even the chemotherapy treatment of DOX in vivo was much less effective compared to the combined treatments. Figure 11a shows the results of the tumor size without any treatment (Saline), with free DOX, with DOX-liposomal system, and with Au-NPs with and without NIR radiation. In Figure 11b, the tumor evolution with and without treatment for breast cancers is shown.

If the cell uptake is affected by the NIR-trigger, the drug release may be affected as well. Campardelli et al. [127] produced gold-PLA nanocarriers (showing the possibility to submit other metals also to this coating [128]), representing interesting possibilities to improve the bioavailability and compliance of these nanoparticles; furthermore, these nanocarriers provided an engineered system capable of transporting a specific payload and triggering it by remote control, obtaining tunable drug release [127].

Concerning the treatments, they can be even more complex. Lee et al. [129] produced PLGA nanoparticles half-covered with an Au layer while the others were loaded with DOX for treatment of an epidermoid cancer line. The in vivo results were very interesting because for direct injection and systematic injection, they obtained a complete regression of the dimensions of the tumor in less than 10 days. For other treatments, such as the only chemotherapy one, the tumor size got doubled in less than one month, even for the same NPs without NIR hyperthermia. The synergic effect of hyperthermia and chemotherapy is accepted now as effective for tumor treatment.

The encouraging results obtained for the in vitro and in vivo trials led to the application of these technologies for clinical trials. In particular, AuroLase Therapy (promoted by Nanospectra Biosciences, Inc., Houston, TX, USA) is one of the first programs on this horizon. It uses gold nanoshells with silica core of ~150 nm, and the first results were published in 2019 for the treatment of prostate tumors on 16 patients (58–79 years old) with NIR photothermal ablation (~810 nm) [130], showing the safety and the feasibility of this technology for the treatment of prostate cancers. In the study, 1 patient did not complete the treatment; for 13 patients out of 15, no evident signs of cancer were detected after one year of treatment.

## 4. Nanoparticles Activated via Pulsed Electric Field

### 4.1. Electroporation and Electrochemotherapy

At present, there is a spreading interest in cancer treatment via a pulsed electric field (PEF) for the promising effects evidenced through this kind of treatment. It can cause aqueous pores in the lipid bilayer due to very short high voltage pulses (generally around the µs and the ms), and it is called electroporation (EP). Even if the mechanism is not perfectly clear [131], the pore generation is probably due to increasing the transmembrane potential due to the electric field [132]. When the transmembrane potential overcomes a determined threshold, the pores start to be generated on the cell membrane [133]. Moreover, the interaction of the field with water molecules, which move because of the field gradient, penetrates in the bilayer of the membrane and accelerates the pore formation process [134].

Depending on the force on the cell membrane, it is possible to study two types of EP: irreversible EP (IEP) or reversible EP (REP) [135,136]. In the case of IEP, the results can be compared to the photothermal ablation because the formation of pores induces cell death [137]. If the pores recover their initial structure so that the survival of the cell is guaranteed, the effects of the electroporation treatment are reversible (REP). The formation of the pores leads to an increase of the permeabilization of the cell membrane, which can be exploited for the diffusion of active substances in the cell. Given the fact that it is usually used as antitumor treatment, it is interesting to study the phenomenon when it is used in combination with chemotherapy, i.e., electrochemotherapy (ECT), similarly to how it was previously done for hyperthermia treatment. Thus, while the hyperthermia combined with the chemotherapy is useful to enhance the transport phenomena and immune response, [86] for the EP, what changes is the structure of the cell membrane, which leads to an enhancement of the transport phenomena. For an electroporated membrane, the mass transport relation is reported in Equation (7) [136].
(7)$$\frac{V}{S}*\frac{dc}{dt}=-D*\frac{z*E*F}{\rho *T}c-D\nabla c$$
where *V* is the volume of the cell, *S* is the surface of the pores, *c* is the active substance concentration, *t* is the time, *D* is the diffusion coefficient, *z* is the electric charge of the molecules or ions, *E* is the local electric field acting on them, *F* is the Faraday constant, *ρ* is the gas constant, and *T* is the temperature. This relation shows that there are two contributions for the transport of molecules [69]. The first one is the diffusive term, D∇c, which is present until the concentration outside and inside the cell is in equilibrium. The second term $D*\frac{z*E*F}{\rho *T}$ is strictly related to the EP, and it remains until the cells recover by the EP. For this reason, it enhances the uptake by cells [138] of low permeant drug, such as bleomycin [139]. Heller et al. [140] studied the combined effects of bleomycin and EP in 34 patients, for a total of 143 tumors, mostly skin tumors and melanomas. After 10 min by the injection of bleomycin, PEF was delivered to the interest zone by the electrode zone. After 12 weeks, the results showed that with only EP or chemotherapy, the results were not good. With ECT, 100% of the patients had a regression for the basal carcinoma; 94% did not show a trace of tumor (CR), whereas the remaining 6% had a regression over than 50% (PR). For the melanoma, the regression was of 98.8% (with 89.3% CR and 9.5% PR), and the others had a 100% regression (80% CR, 20% PR). These results were interesting because they showed a synergic effect of the combination of the two treatments. Moreover, this strategy showed the efficacy of the bleomycin, which usually is not effective for the treatment of solid tumors [140]. It is a very cytotoxic agent, but it is limited by the fact that it is not able to overcome the plasma membrane [141]. In this way, electroporation opened a new way of drugs potentially usable for cancer treatments. Gibot et al. [142] in 2013 proved the efficacy of ECT combined with bleomycin, cisplatin, and DOX treatment for the treatment of a colorectal carcinoma in a multicellular tumor spheroid. It was proven that the ECT-bleomycin treatment led to 10 times more apoptosis than the bleomycin alone. The same phenomenon happened for the cisplatin, whereas the ECT-DOX showed about 5 times more apoptosis than the DOX one. Moreover, it was observed that there was a maximum decrease of the apoptosis cells (which were in the external layers of the tumor) that went into the core. The electroporator is generally constituted by three main parts: the generator, the electrodes, and the AccuSync device. The generator and the electrodes instruments are necessary for the electroporation, whereas the AccuSync device is used to deliver a PEF synchronized with the patient electrocardiogram [135,143]. The electrodes are crucial elements because their configuration substantially affects the electric field generated [144]; for this reason, different types of electrodes were produced. For example, they can be regular plates placed on the side of tumor in order to generate a homogenous electric field in the carcinoma. Moreover, if the interested region is subcutaneous, the parallel plates may not guarantee an adequate electric field. For example, they can be replaced by needle electrodes that show better performance and an easier application for some types of tumors [145]. However, the most relevant problem of electroporation is that it is applicable for easily accessible tumors, mostly external, as well as the potential cell damage induced by using high voltage.

### 4.2. Combination of Electroporation and Nanoparticles

NPs combined with electroporation can considerably enhance EP performance, as they have beneficial effects on the classical drawbacks of the treatment. Zu et al. [146] studied the differences in the classical EP inducted by the electroporation for mammalian and leukemia cells in vitro by using gold NPs. To pore the cell, it is necessary to have a certain potential difference between two edges of the cell membrane, and it is found using this equation:(8)$$\Delta V_{m}=\frac{3}{2}*E_{ext}*r*\cos\theta$$
where *E* is the electric field strength, *r* is the radius of the cell, and *θ* is the angle between the electric field and the membrane [146]. The necessary voltage delivered by the electroporator, however, is always more than the voltage required for the breakdown of the cell membrane. This happens because the cell surrounding environment works as a resistance for the two electrodes, as reported in Figure 9. This is one of the reasons that causes cell damage: It is necessary work with a higher voltage than that valued necessary. Putting inside gold NPs, characterized by a high conductivity (~4.5 × 10^6^ S/m), the resistance through the environment is considerably decreased. Figure 12 presents the scheme of this NP functionality.

The results show that, as predicted, it is possible to work with a lower voltage, which increases the cell viability, in turn obtaining high transfection efficiency (analyzed monitoring fluorescent molecules transfection in the cells). However, another important difference is reported regarding the genesis of pores. The NPs act as microelectrodes that polarize the cell by different sides. This is quite different from the standard electroporation, which causes mainly two large breakdowns. The same results were obtained also for the leukemia cells. In particular, the greatest results are obtained on the cell viability, which seems to be higher than other innovative EP treatment in literature [147]. Continuing this path, Huang et al. [148] produced Au-NPs coated with polyethyleneimine (PEI) and functionalized with siRNA or DNA plasmids (polyplex/Au-NPs). The PEI was fixed, exploiting electrostatic interactions; this was considered necessary in order to avoid degradation, during the EP, which leads to freeing cationic molecules. The results show a considerable improvement of the system Polyplex/Au-NPs compared to the free polyplex, both in terms of cell viability and in terms of transfection efficiency. This is probably due to the excellent biocompatibility of the Au-NPs, to the stability that the NPs give to the polyplex, and to the synergistic effect of EP with the NPs mentioned before. The scientific interest in this field is now spreading, and many researchers are working to investigate these phenomena in depth [149].

The first clinical trials for electrochemotherapy started in 1991 [150], and over time it became clinically accepted for several types of tumors and treatments [151,152]. However, the combination of these stimulations with NPs is still in development, and at the moment to the best of authors’ knowledge, there is no information on clinical trials that are already reported in literature.

## 5. Transdermal Delivery Systems Stimuli-Responsive

### 5.1. Nanofiber and Nanoparticle System: Properties

In the last decade, the interest in the combination of nanoparticles with nanofibers has increased. Among all the existent processes able to produce nanofibers (NFs), electrospinning is probably the most interesting for biomedical applications. In the electrospinning process, the polymeric solution is fed to the injector, and in exploiting the electric field generated between the injector and the collector, the generated stresses go to thin the polymeric jet, reducing the dimension of the produced fibers [153,154]. Moreover, it is possible to produce NFs of very different materials with very different properties. In Figure 13, a schematic illustration of the electrospinning process and a SEM image of a polycaprolactone (PCL) membrane are shown.

To produce composite materials, the NPs can be loaded both after the production of NFs (indirect method, for example, immersing the membrane in the colloidal solution) and during the electrospinning process (direct method, for example, dissolving the NPs in the polymeric solution), which is usually preferred since it allows for easy control of the amount loaded in the NFs [155]. Following this path, now the scientific community is showing interest in the encapsulation of the NPs in the NFs to make nanocomposite-materials stimuli-responsive. Before having NFs express the stimuli-responsive properties, it is relevant to underline that different types of NPs lead to different characteristics of the NFs in terms of morphology, mechanic properties, and general performance. For example, Jin et al. [156] produced polyvinyl-alcohol (PVA) NFs via electrospinning loaded with different sizes of silica (SiO_2_) NPs. For a higher diameter (910 nm) of the NPs, the NFs presented a considerable change of morphology compared to the ones without NPs. The structure formed was similar to a necklace, where the polymer linked each particle. By decreasing the NPs size (143 nm), the morphology was found newly different compared to the previous NFs. In this last case, an aggregation of NPs, which created a visible crunch in the filament morphology, was observed. Finally, in reducing the NP diameter (<80 nm), it is possible to encapsulate NPs efficiently in the polymeric matrix without showing superficial modifications, making them bounded in the filament [157,158]. However, the NPs characteristics can be exploited also to modify the morphology of the NFs. For example, Hu et al. [159] produced poly(lactic-co-glycolide) (PLGA) loaded with magnetic NPs using an electrospinning system in the presence of a magnetic field (generated by a permanent magnet). The SEM analysis proved the alignment of the nanofibers caused by the interactions between the magnetic NPs and the magnet, demonstrating that the nanocomposite materials allow for changing the morphological properties. The mechanical properties are sensitive to the loading of NPs as well. Lee et al. [160] produced PVA nanofibers loaded with different percentage of montmorillonite, proving that the filler affects the mechanical properties of the material, which in turn increased the tensile strength and modified many parameters [161]. Similarly, the research performed by McKeon-Fischer et al. [162] and Wei et al. [163], which concerns the changes in mechanical properties of an electrospun bio-matrix charged with Au-NPs and Fe_3_O_4_ NPs, showed very similar results. In this context, considerations on biomedical performance are even more interesting. In fact, McKeon-Fischer et al. [162] also highlighted the lack of toxicity of the Au-NP/matrix system, which was used as a scaffold for the healing process of the skeletal muscle. For cancer treatment, Guadagno et al. [164] proved that the use of PCL membranes loaded with functionalized magnetite NPs (PCL-Fe_3_O_4_) was particularly effective for the treatment of melanomas and for uterine cervix cancer even with no drug and no hyperthermia-induced treatment. In evaluating the viability of the cells, the results showed that for HeLa cells, the maximum abatement of the cell viability was reported for the system combined of PCL-Fe_3_O_4_, with almost a 35% cell viability recorded at the end of the treatment. For the treatment of melanoma, however, one very interesting result was recorded. The PCL membrane that was unloaded determined a slight increase in the cell viability compared to the control ones, but by loading the nanofibers of the membranes with magnetic NPs, it is possible to obtain results competitive with the results related to the single magnetic NPs, with a decrease of the cell viability up until 35%. The results relative to the A375 cell viability are reported in Figure 14.

Other types of analysis were done to investigate the behavior of nanocomposite-NFs. The literature highlights how the biodegradation of the polymeric matrix generally occurs before for the nanocomposite material. In particular, the higher the number of NPs is, the faster the degradation of the biopolymer [157,165,166,167].

### 5.2. Stimuli-Responsive Nanocomposite Materials

Currently, even if several studies report the performance of the nanocomposite systems if stimulated by external sources, very few studies deal with the electromagnetic characteristics of the materials and the interaction among the radiation source and the nanocomposite materials. Electrospun systems are characterized by high porosity, and for this reason, electrospun membranes obviously show different electromagnetic characteristics compared to the bulk materials [168]. Moreover, with the inclusions of nanoparticles, the characteristics can keep changing not only in terms of, for example, the electric conductivity and dielectric permittivity, but also as regard the transmitted and reflected energy for waves interaction through the nanocomposite material containing the filler [169,170]. The EP is perfectly adaptable to applications conjugated with the drug delivery via electrospun membrane. Very recent studies started matching these two systems to investigate their potentialities if applied together. In 2018, Wang et al. [171] loaded hydroxycamphothecin (HCPT) on an electrospun membrane to treat VX2 subcutaneous cancer of rabbits together after being ablated through IEP. The results were promising, since they showed a tumor inhibition rate more than 80%, while the system with only electroporation or the one only with drug membrane manifested a tumor inhibition rate of 62% and 38%, respectively. Moreover, the tumor mass was almost around 30% compared to the control tumor mass, whereas for the other treatments the mass was about 40% and 62%, respectively. However, this highly cutting-edge system is now in development and promises to be highly interesting and with high potentiality, given its usability and the matching between two clinical-used systems (i.e., transdermal delivery and electroporation).

However, as regard the hyperthermic potentialities, certain NPs such as Au-NPs, for example, present an absorbance peak that is very similar even after being encapsulated in nanofibers during the electrospinning process [172]. This result suggests that the photothermal activity is conserved, as experimentally evidenced. In this last context, Cheng et al. [173] produced an electrospun membrane of poly(lactic-co-glycolic acid) (PLGA) and PLA-b-PEG loaded with gold nanorods. The nanorods were functionalized with PEG, with a dimension of 52 nm and a diameter of 10 nm. First of all, the Au-nanorods release was evaluated. In two hours, 30% of Au-Nanorods were released, while for the remaining 70%, it was necessary to wait three months, showing that the usage of a membrane is effective for the release control. Moreover, the cell uptake for a lung cancer cell line was almost 80%. The cell line was stimulated via NIR radiation, causing hyperthermia up to 42 °C, and not only was a high rate of cell death guaranteed for the lung carcinoma line, but by testing also a human bronchial Epithelial cell line, the selectivity of the treatment was proven, guaranteeing a high vitality for the healthy cells, a high cancerous cell death, and a selective inhibition of the cell proliferation (it decreases only for the cancerous cell line). Similar results were obtained with magnetic induced hyperthermia treatment. As mentioned before, magnetite-based NPs can be loaded on or into electrospun fibers during the process [174] and at the end of the process [175]. It was found that by increasing the number of NPs, the heating rate is generally higher, as well as the maximum reachable temperature; secondly, the increase in temperature can lead to modifications on the polymer structure: For example, in case of PCL, whose melting point is 61 °C, by exploiting adequately the hyperthermia, it is possible to induce the melting of the matrix. Eventually, by encapsulating the NPs in the matrix, it is possible to slow down the heating rate considerably, making the process more controllable [176]. Lin et al. [176] in 2012 produced chitosan (CS) NFs that were then treated via two ways for the loading of NPs, i.e., immerged in a magnetic nanoparticle solution or immerged in a Fe^2+^/Fe^3+^ solution followed by a coprecipitation of the NPs. The in vitro results were evaluated for the treatment of a colon adenocarcinoma and on a lung fibroblast line, showing that the Fe_3_O_4_ NPs have no cytotoxic effect on that tumor line, but the hyperthermia treatment is very effective for the single NPs, and similar results were obtained for the systems NFs-NPs with magnetic hyperthermia, inducing a cell death around the 70%. All these results, also obtained with the combination of other systems to generate hyperthermia [177] reasonably, led to a very new generated interest toward these systems, especially if combined with the chemotherapy. Niyama et al. [178] in 2019 produced a PCL membrane loaded with magnetic nanoparticles and paclitaxel (PXL) to be implanted after surgical resection in the pleural cavity. In this case, the membrane was composed of a PCL matrix and two filler: the magnetic NPs and PXL. The results show that the drug release was not improved by the hyperthermia, and the cell viability and the tumor size were considerably influenced. In particular, in vitro tests showed that in only one day of treatment, the PXL-NPs-hyperthermia membrane system was sensibly more effective with respect to the membrane without NPs and stimulus, inducing a cell death of 42% in 24 h, compared to the chemotherapeutic treatment on membrane having integrated only PXL, which determines cell death only of 8%. The promising results obtained in vitro were confirmed in vivo when treating the same tumor line in mice. In particular, with the stimulation of the alternate magnetic field of 15 min per week, for a total duration of 8 weeks, the evolution of the tumor mass in mice was studied. The study highlighted that the treatment was more effective via transdermal delivery than via injection (because of the short half-life of the chemotherapeutic agent alone), and the combined treatment brought a 87% reduction of the tumor mass, much higher than any other treatment. Similarly, Park et al. [179] produced a PCL membrane loaded with gold nanocages (receptive to NIR radiation), DOX (chemotherapy agent), and fatty acids for the treatment of a breast cancer cell line. The use of fatty acids has proven to be crucial for a sustainable release. As previously observed, PTX release was not influenced by the magnetic hyperthermia in the PCL membrane. The fatty acids are in fact a phase-changeable agent, which melts above 39 °C and goes out of the membrane. In this way, small pores are generated, and the superficial area is increased. In evaluating the release, it is possible to observe that, without NIR radiation, the drug release is very slow. After 5 min of NIR radiation, the rate for the membrane without fatty acids was not sensibly changed, whereas increasing the amount of fatty acids, the kinetic of release changed sensibly. Cell viability confirms these results: Without NIR radiation, the cell viability is above 90%; because there are no effects of the hyperthermia and very low effects of the chemotherapy, after one treatment it decreases to 57% (without fatty acid is 77%) and after four cycles of treatment the viability decrease to 4%. In Figure 15, the genesis of pores after NIR and the release profile are reported.

The abovementioned changes are irreversible: Once the fatty acids have abandoned the membrane, the pores are generated and the release rate results changed. Li et al. [180] produced a poly(N-isopropylacrylamide) (PNIPAM) membrane loaded with DOX-silica-coated Au-nanorods with the polyhedral oligomeric silsesquioxane (POSS) as a crosslinking agent for a sustainable kinetic release that is very interesting. PNIPAM is a thermosensitive material that can reversibly change hydrophilic/swelling state to a hydrophobic/deswelling state when moving above a critical temperature, known as lower critical solution temperature (LCST). For the PNIPAM, it is usually around 30–34 °C, but it can be shifted through a copolymerization (which, in this case, through the POSS) [181]. In evaluating the photothermal effect on volume phase transition of the material, it was possible to notice the great decrease in volume, about 83%, when irradiated by NIR radiation. Similar changes were reported from hydrophilic behavior to hydrophobic behavior. These structural changes determine a different release kinetic of DOX for the material, i.e., when not irradiated and when irradiated. In this way, it is possible to temporarily modify the material properties and then return to the non-irradiated ones. However, electrospinning is a process particularly versatile, which allows for the production of bilayer structure through the coaxial-electrospinning process. To optimize the release kinetics, the employment of sheath-core membranes is a very interesting strategy. Zheng et al. [182] produced a polymeric membrane in which the core is composed of polyethyleneimine (PEI) and a DNA plasmid that has to be transfected in the cells, whereas the sheath is composed by PLA/Gelatin with Au-nanorods (responsive filler). This study evaluated the gene transfection and the release through a coaxial system stimuli-responsive for the treatment of a fibroblast cell line. As regard the release, without NIR radiation after 90 s, less than 1.5 ng of the gene was detected, whereas with NIR radiation the same system released about 80 ng (about 50 times higher). Moreover, when stopping the irradiation, the release of the plasmid stopped, suggesting that it may be used for at least three cycles. It is possible that not only the rise of temperature enhances the transport phenomena affecting the permeability, but also the disruption of the cell membrane near to the mat [183]. The transfection was evaluated through fluorescent protein, and the nanocomposite system under NIR shows a transfection efficiency at least 4 times higher than the system without radiation and without Au-NPs and guarantees cell viability higher than 90%.

## 6. Future Perspectives and Conclusions

In summary, the scientific research highlighted that the future nanomedicine lies in the use of innovative, smart materials able to react to accurately chosen stimuli that are able to temporarily modify the properties of the materials. In order to obtain these results, the scientific community is now focused on the detection of susceptible elements with corresponding stimuli. The stimuli studied for this type of application are mainly electromagnetic, and they have a relevant impact on the stimulation and permeabilization of the cells (e.g., EP) or on the thermal characteristics of the systems (e.g., NIR radiation and AMF). The continuous development of specific NPs or NPs-based systems is leading to more effective treatment able to combine proven therapies, i.e., pharmacological and hyperthermic ones, able to enhance the immune response and the uptake of the drug by the cells. If these biological aspects can sensitively enhance the performance of the drug, even more interesting aspects can obtained by the integration of these stimuli-responsive domains in transdermal delivery systems. The analysis of the papers presented in this work shows, on the one hand, the characteristics of nanocomposite electrospun systems, and on the other hand, the possibility to modify their morphology and structure to adapt the system to the planned stimulus. As seen before [179], changes in morphology lead to important variations in the release kinetics. These modifications, depending on the design of the system, can be irreversible, leading to the development of a completely new material without the possibility to go back to the first morphology, or reversible, in the case where it is possible to go back to the first morphology and therefore to the previous characteristics. The first type of material brings immediately interestingly applications in which it is possible to modify the characteristics of the system (as reported in Figure 13), obtaining systems in which it is possible to control the behavior and activate a new release kinetic under specific conditions. The pros of these systems lie in actively controlling the morphology of the transdermal delivery system, and it is possible to obtain these systems theoretically with each type of material, as the PCL reported by Park et al. [179]. The reversible systems, as reported, can be theoretically even more performing, since it is possible to control in time the morphology of the system until the end-use of the material. Ideally, it may be compared to the possibility of having a button for controlling the morphology of the system. Obviously, with these perspectives, the materials have to be chosen very accurately, for example, thermosensitive polymers such as the PNIPAM [180]. The imposed condition allows for organizing the structure and the morphology differently as long as the stimulus is active. This vision will probably lead to massive development of these new types of systems in the next future.

## Figures and Tables

**Figure 1 nanomaterials-11-00848-f001:**
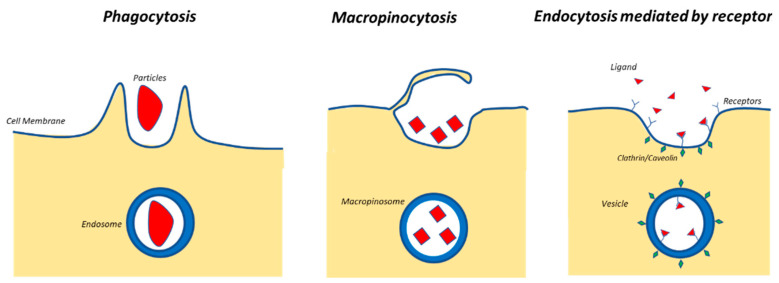
Main types of endocytosis of nanoparticles (NPs).

**Figure 2 nanomaterials-11-00848-f002:**
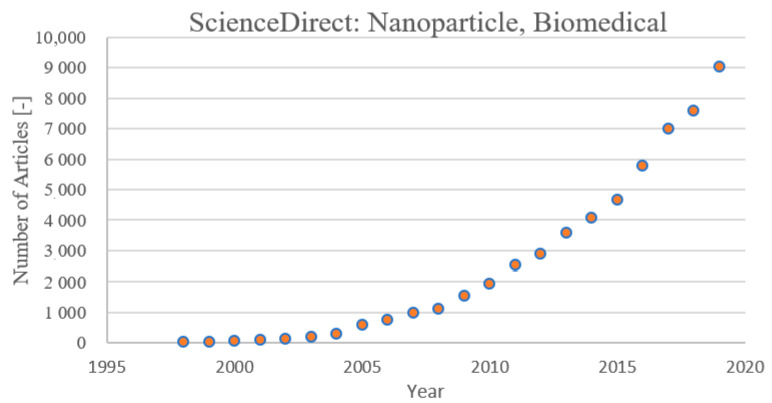
Publications per year on ScienceDirect from 1997 to 2019 on NP application in the biomedical field.

**Figure 3 nanomaterials-11-00848-f003:**
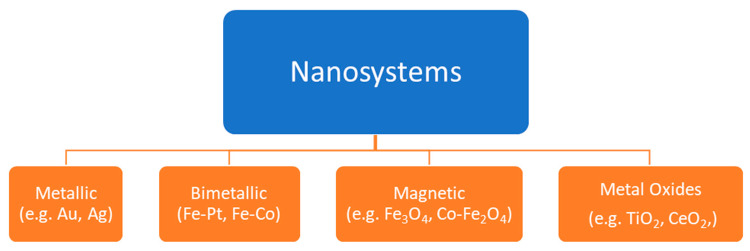
Classification of nanoparticles.

**Figure 4 nanomaterials-11-00848-f004:**
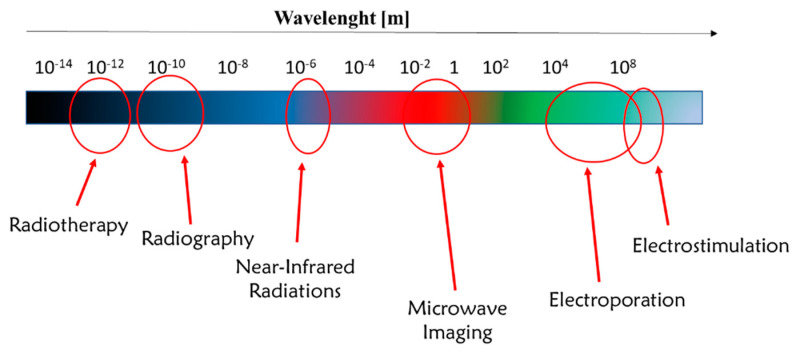
Applications of electromagnetic radiations in medicine.

**Figure 5 nanomaterials-11-00848-f005:**
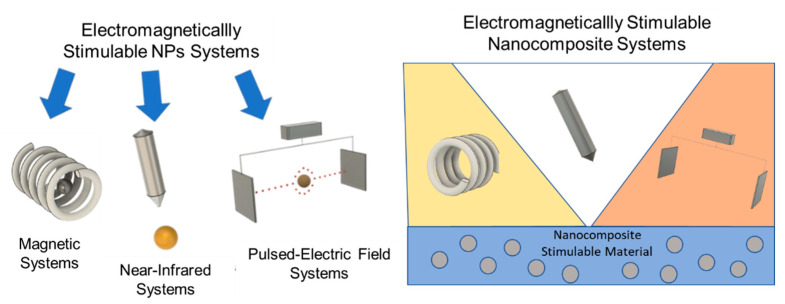
Electromagnetically stimulable systems.

**Figure 6 nanomaterials-11-00848-f006:**
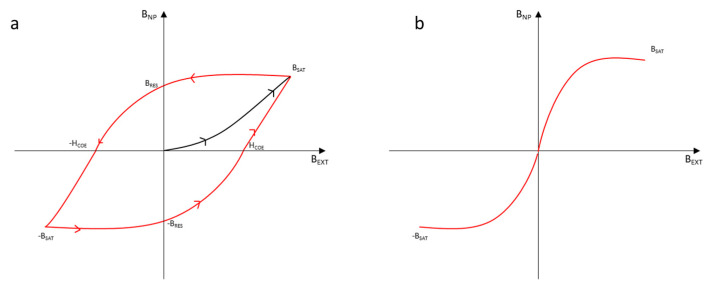
Route of the magnetic field provided by a NP (B_NP_) depending on the external magnetic field (B_ext_) applied for a ferromagnetic material (**a**) and a superparamagnetic material (**b**).

**Figure 7 nanomaterials-11-00848-f007:**
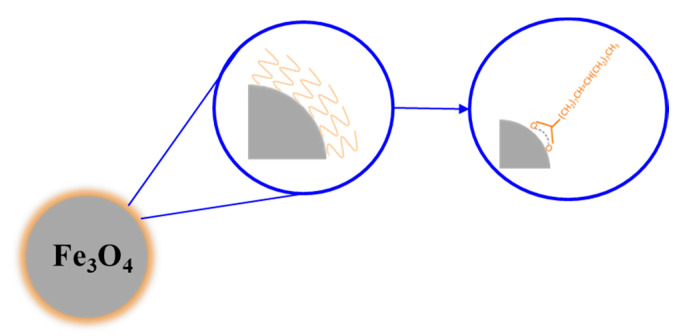
Magnetite NP functionalized with oleic acid.

**Figure 8 nanomaterials-11-00848-f008:**
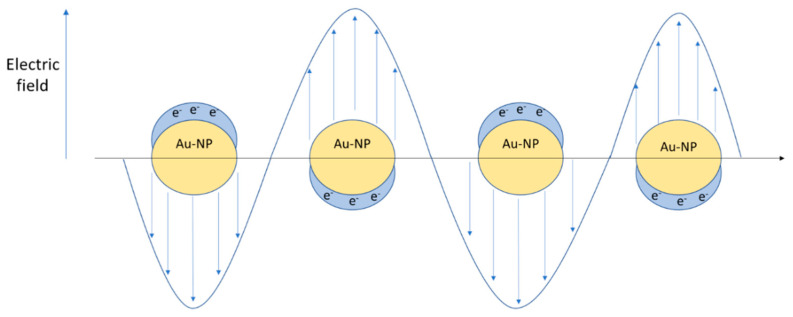
Localized surface plasmon resonance (LSPR) induced by external radiation.

**Figure 9 nanomaterials-11-00848-f009:**
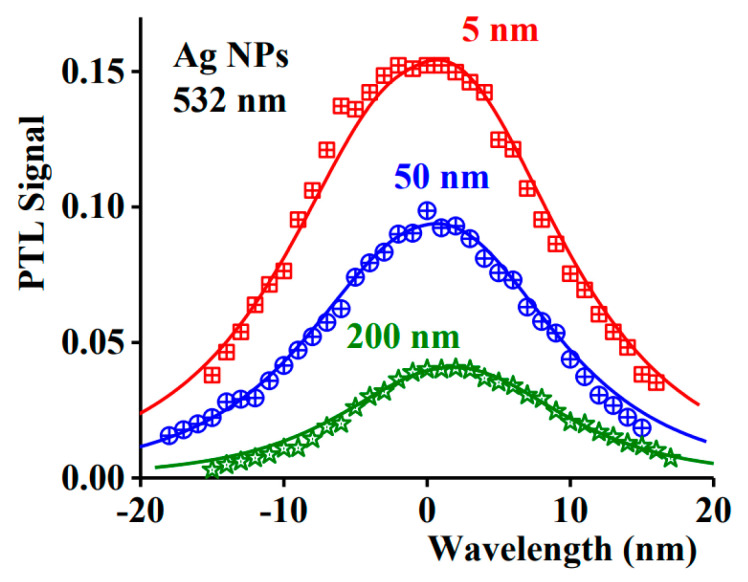
Photothermal lens signal vs. wavelength for various size of NPs (Reproduced from [103] with permission from Elsevier, 2021).

**Figure 10 nanomaterials-11-00848-f010:**
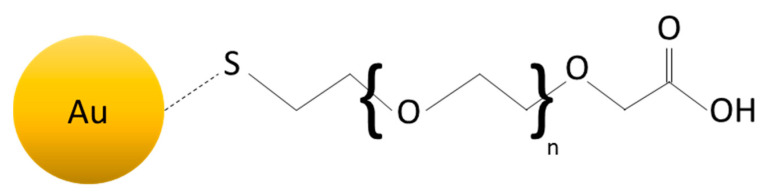
Example of Au-NP functionalized with thiol-carboxyl group.

**Figure 11 nanomaterials-11-00848-f011:**
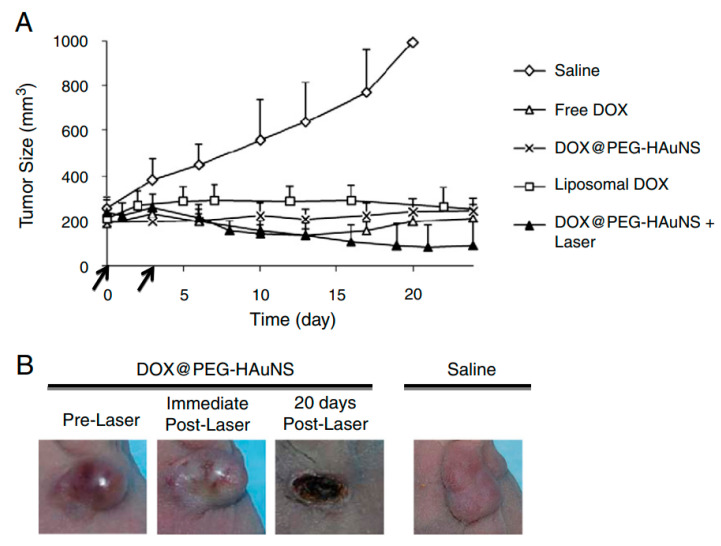
(**A**) Development of tumor size for each treatment depending on the size; (**B**) tumor evolution for doxorubicin (DOX)/Au-NPs with near-infrared (NIR) radiation and without any treatment (Reproduced from [126] with permission from Elsevier, 2021).

**Figure 12 nanomaterials-11-00848-f012:**
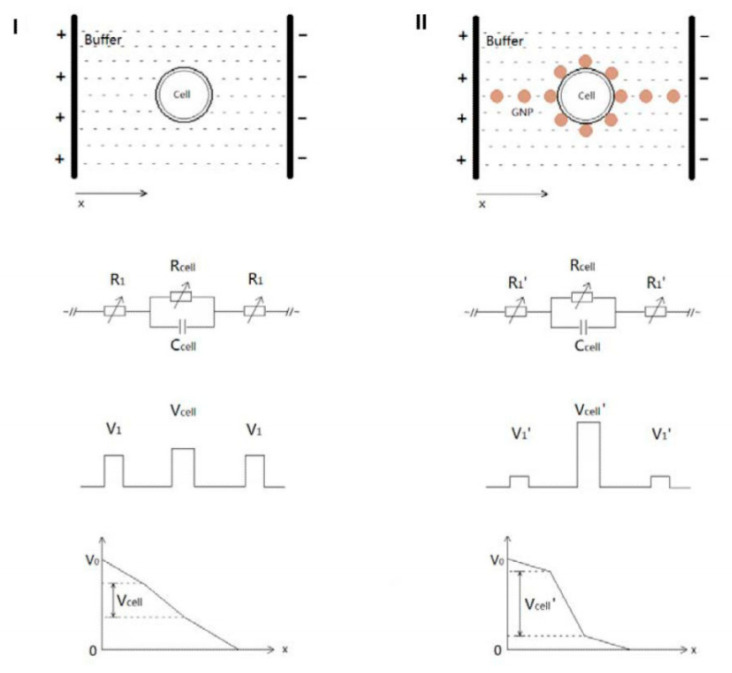
Differential of potential in a biological environment without metallic NPs (**left**) and with NPs (**right**). Reprinted from “Gold nanoparticles enhanced electroporation for mammalian cell transfection.” *J. Biomed*. *Nanotechnol*. 2014, 10, 982–992, Figure 1” [146].

**Figure 13 nanomaterials-11-00848-f013:**
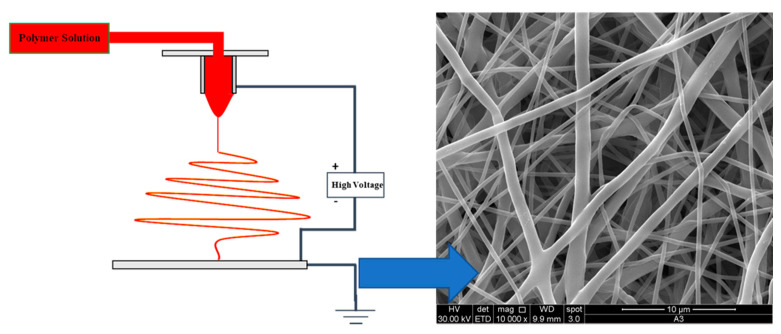
Electrospinning process (**left**) and SEM image of a polycaprolactone (PCL) membrane (**right**).

**Figure 14 nanomaterials-11-00848-f014:**
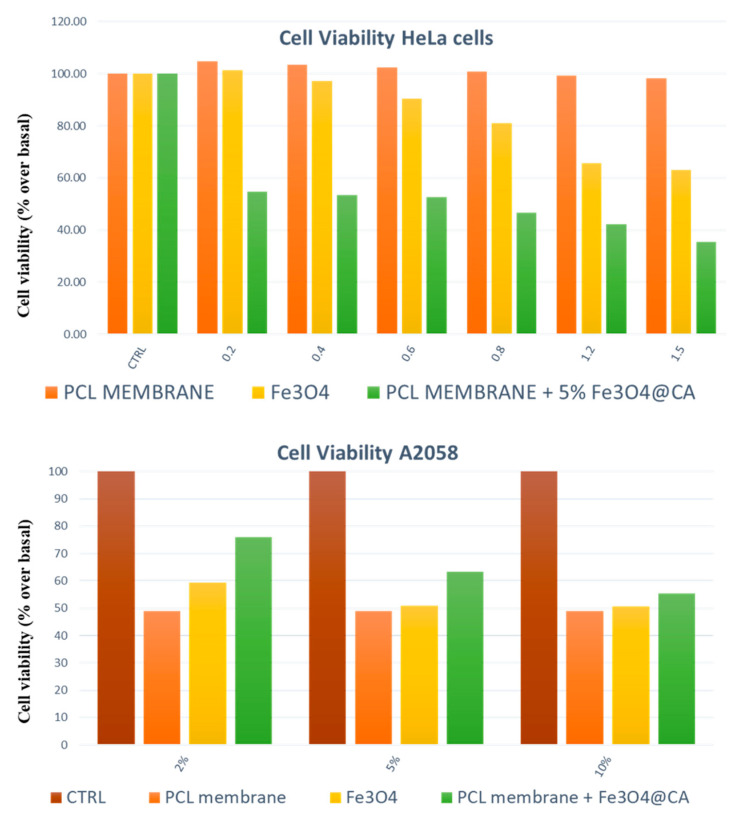
Cells viability for cervix uterine cancer (**up**) and melanoma (**down**) cell lines.

**Figure 15 nanomaterials-11-00848-f015:**
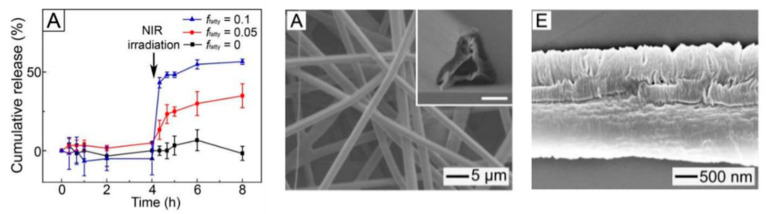
(**left side**) Cumulative release of DOX-gold-nanocages-membrane for various fatty acids values; (**middle**) SEM of the membrane with 0.05 fatty acids before NIR radiation; (**right side**) SEM of the membrane with 0.05 fatty acids after NIR radiation. Reprinted from “Gold nanocage-incorporated poly(ε-caprolactone) (PCL) fibers for chemophotothermal synergistic cancer therapy.” *Pharmaceutics* 2019, 11, 60, Figure 6 and Figure 7 [179].

**Table 1 nanomaterials-11-00848-t001:** Table of the contents.

Type of System	Type of Stimulus	Topics
Stimulable nanoparticles	Magnetically	Activation mechanism stimulus (NPs)
		Characteristics of NPs
		Preclinical and clinical studies
	NIR radiation	Activation mechanism stimulus (NPs)
		Characteristics of NPs
		Preclinical and clinical studies
	Pulsed electric field	Activation mechanism EP and interaction stimulus (NPs)
		Modulation of relevant parameters
		Preclinical tests and potentialities
Stimulable nanocomposite	Magnetic NIR radiation pulsed electric field	Characteristics of nanocomposite electrospun materials
		Stimulability of nanocomposite electrospun membranes
		Preclinical tests and potentialities of electrospun membranes
**Conclusions**		Potentialities and perspectives

**Table 2 nanomaterials-11-00848-t002:** Effect of the physicochemical properties of NPs on biological performance.

Physicochemical Parameter of NPs	Effect
Shape	Cytotoxicity [50]
	Uptake [51]
	Distribution [52]
Size	Uptake [41]
	Distribution [53]
	Cytotoxicity [41]
Surface charge	Uptake [49]
	Cytotoxicity [49]
	Distribution [54]

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
