# Peer review of "Electromagnetically Stimuli-Responsive Nanoparticles-Based Systems for Biomedical Applications: Recent Advances and Future Perspectives"

_nanomaterials, 2021, doi:10.3390/nano11040848_

Round 1
Reviewer 1 Report
No revision needed.Author Response
The authors thank the reviewer for his/her effort during the revision process.

Reviewer 2 Report
The authors politely answered all my questions and comments. Therefore, the manuscript will be suitable for this journal.
Author Response
The authors thank the reviewer for his/her availability to review this manuscript.

Reviewer 3 Report
I am not quite satisfied with the Authors' efforts to improve their text. My further comments:
- Not all metallic NPs will respond to NIR in a favourable way. Some more elaboration needed.
- Eq. 3, what is µ0? Is it the same as µ in Eq.2?
- P. 8. The Authors should discuss in more details the hydrophilic/hydrophobic balance requirements for the NPs both for the suspension stability, as well as drug carriers.
- P. 12. Discussing the apoptosis and necrosis for NIR temperature rise, the same concern should be stressed in the case of magnetic hyperthermia.
- Eq. 17. The units throughout this equation are inconsistent. The left-hand side is mol/cm2t and so is the far right-hand side, but the middle term (after the equilibrium mark) has both T and density in the denominator, having energy in the nominator. Therefore, either it is a typo, or propagation of erroneous equation from literature
Author Response
Author's Reply to the Review Report (Reviewer 3)
The authors thank the reviewer for the useful suggestions and questions. Below the authors have answered per point to all the comments of the reviewer.
Point 1
- Not all metallic NPs will respond to NIR in a favourable way. Some more elaboration needed.
Answer to the Reviewer comments
The authors thank the reviewer for the comment. As reported in Pag. 10-11, NIR response of metallic NPs is based on the response of free-moving electrons. For this reason, metallic NPs with a great quantity of free moving electrons (e.g, Au NPs. Ag NPs) are particularly suitable. [Searching for better plasmonic materials, Laser Photonics Rev. 4, No. 6 (2010), 795-808]
Furthermore, the design of the NPs can greatly affect their response to stimuli. Mainly, by varying the size and shape of NPs it is possible to affect the radiation absorbance and, so, the heat dissipated. But if the resonance wavelenght of the NPs is not in the NIR region it is possible to shift the absorbance peak my modifying the designing of NPs (for example, forming nanoshells).
In the second revision of this manuscript a brief discussion has been added to better describe possible materials to use with NIR stimulation.
Changes in the text
Pag. 12: In synthesis, a suitable NIR-responsive NP should show a relevant absorption in the NIR region that can be converted in heat. The most used materials in these sense are noble metal nanomaterials because of the strong plasmon resonance, even if now also other type of materials are becoming of interests for being studied (e.g. graphene-based materials). [111,112]
- Chen, Y.W.; Su, Y.L.; Hu, S.H.; Chen, S.Y. Functionalized graphene nanocomposites for enhancing photothermal therapy in tumor treatment. Adv. Drug Deliv. Rev. 2016, 105, 190–204, doi:10.1016/j.addr.2016.05.022.
- Lv, Z.; He, S.; Wang, Y.; Zhu, X. Noble Metal Nanomaterials for NIR‐Triggered Photothermal Therapy in Cancer. Adv. Healthc. Mater. 2021, 10, 2001806, doi:10.1002/adhm.202001806.
Point 2
- Eq. 3, what is µ0? Is it the same as µ in Eq.2?
Answer to the Reviewer comments
The authors thank the reviewer for this very useful comment. In Eq. 2 µ is the viscosity of the system, whereas in Eq. 3 µ0 is the magnetic field constant. The authors has specified the meaning of µ0 in the text at Pag. 6 below Eq. 3.
Changes in the text
Pag. 6: where P is the heat dissipation, f is the frequency of the external magnetic field, H is the strength of the external magnetic field, µ0 is the magnetic field constant and ν is the imaginary part of the magnetic susceptibility. [39]
Point 3
- 8. The Authors should discuss in more details the hydrophilic/hydrophobic balance requirements for the NPs both for the suspension stability, as well as drug carriers.
Answer to the Reviewer comments
The authors thank the reviewer for the useful comment. The authors have improved the manuscript by briefly discussing of the relevance of the hydrophilic/hydrophobic balance. The changes in the text are reported below.
Changes in the text
Pag. 8: The balance between hydrophilicity and hydrophobicity for the NPs is particularly challenging. In fact, on the one hand the hydrophilicity allows a good dispersion in serum or in water, avoiding aggregation. For the water dispersibility generally it is possible to attach hydrophilic functional groups to the ligands (e.g. carboxylic acids, sulfonic acids, PEG, etc.). [73] On the other hand the hydrophobicity enhances the interactions among the NPs and the cell membrane and, so, their uptake. [74,75] Given the fact that surface charge affects the cell uptake (generally positive surface charge increases the uptake), the use of cationic ligands was one of the main ways used to overcome the cell membrane. However, now the research is also going toward the development cutting-edge NPs with hydrophobic/hydrophilic faces (Janus type) or hydrophilic/hydrophobic triggerable properties. [73]
- Kobayashi, K.; Wei, J.; Iida, R.; Ijiro, K.; Niikura, K. Surface engineering of nanoparticles for therapeutic applications. Polym. J. 2014, 46, 460–468, doi:10.1038/pj.2014.40.
- Fratoddi, I. Hydrophobic and hydrophilic au and ag nanoparticles. Breakthroughs and perspectives. Nanomaterials 2018, 8, 11.
- Honary, S.; Zahir, F. Effect of zeta potential on the properties of nano-drug delivery systems - A review (Part 1). Trop. J. Pharm. Res. 2013, 12, 255–264, doi:10.4314/tjpr.v12i2.19.
Point 4
- 12. Discussing the apoptosis and necrosis for NIR temperature rise, the same concern should be stressed in the case of magnetic hyperthermia.
Answer to the Reviewer comments
The authors thank the reviewer for the useful comment. The authors have added a description in the manuscript of the effect of temperature rise induced by magnetic hyperthermia, as done for the NIR-induced hyperthermia. The changes in the text are reported below.
Changes in the text
Pag. 9: Magnetic hyperthermia can cause protein denaturation and DNA damage of the cells. It can enhance the immune response and the drug perfusion within the tumor. [80,81] Cell death via magnetic hyperthermia can be due to apoptosis or necrosis. However causing cell death via apoptosis is preferred than necrosis since necrosis can provoke inflammation or metastasis. [82]
Pag. 13: The hyperthermia caused by NIR radiation has several physiological effects on the cells. Firstly, cancer’s vessels and cell membrane permeability increase thanks to the rise of temperature, which causes a more efficient drug uptake. However, there are also other effects directly inside the cells: the temperature increase also causes protein denaturation and DNA damage, as also detected in the case of magnetic hyperthermia [123]
- Chang, D.; Lim, M.; Goos, J.A.C.M.; Qiao, R.; Ng, Y.Y.; Mansfeld, F.M.; Jackson, M.; Davis, T.P.; Kavallaris, M. Biologically targeted magnetic hyperthermia: Potential and limitations. Front. Pharmacol. 2018, 9, 831, doi:10.3389/fphar.2018.00831.
- Cellai, F.; Munnia, A.; Viti, J.; Doumett, S.; Ravagli, C.; Ceni, E.; Mello, T.; Polvani, S.; Giese, R.W.; Baldi, G.; et al. Magnetic hyperthermia and oxidative damage to dna of human hepatocarcinoma cells. Int. J. Mol. Sci. 2017, 18, doi:10.3390/ijms18050939.
- Moise, S.; Byrne, J.M.; El Haj, A.J.; Telling, N.D. The potential of magnetic hyperthermia for triggering the differentiation of cancer cells. Nanoscale 2018, 10, 20519–20525, doi:10.1039/c8nr05946b.
Point 5
- 17. The units throughout this equation are inconsistent. The left-hand side is mol/cm2t and so is the far right-hand side, but the middle term (after the equilibrium mark) has both T and density in the denominator, having energy in the nominator. Therefore, either it is a typo, or propagation of erroneous equation from literature
Answer to the Reviewer comments
The authors thank the reviewer for the useful comment. Since there is not Eq. 17 in the manuscript, the authors presume that the reviewer was referring to Eq. 7. Regarding Eq. 7, the authors thank the reviewer for the useful comment, actually by mistake ρ was defined as density. Now in the manuscript ρ has been defined correctly as the gas constant. Thank you very much for the useful comment.
Changes in the text
(7)
Where V is the volume of the cell, S is the surface of the pores, c is the active substance concentration, t is the time, D is the diffusion coefficient, z is the electric charge of the molecules or ions, E is the local electric field acting on them, F is the Faraday constant, ρ is the gas constant and T is the temperature.
